# GRU-ODE-Bayes: Continuous modeling of sporadically-observed time series

**Edward De Brouwer**[*][†]
ESAT-STADIUS
KU LEUVEN
Leuven, 3001, Belgium
edward.debrouwer@esat.kuleuven.be

**Jaak Simm**[*]
ESAT-STADIUS
KU LEUVEN
Leuven, 3001, Belgium
jaak.simm@esat.kuleuven.be

**Adam Arany**
ESAT-STADIUS
KU LEUVEN
Leuven, 3001, Belgium
adam.arany@esat.kuleuven.be

**Yves Moreau**
ESAT-STADIUS
KU LEUVEN
Leuven, 3001, Belgium
moreau@esat.kuleuven.be

## Abstract

Modeling real-world multidimensional time series can be particularly challenging when these are *sporadically* observed (*i.e.*, sampling is irregular both in time and across dimensions)—such as in the case of clinical patient data. To address these challenges, we propose (1) a continuous-time version of the Gated Recurrent Unit, building upon the recent Neural Ordinary Differential Equations (Chen et al., 2018), and (2) a Bayesian update network that processes the sporadic observations. We bring these two ideas together in our GRU-ODE-Bayes method. We then demonstrate that the proposed method encodes a continuity prior for the latent process and that it can exactly represent the Fokker-Planck dynamics of complex processes driven by a multidimensional stochastic differential equation. Additionally, empirical evaluation shows that our method outperforms the state of the art on both synthetic data and real-world data with applications in healthcare and climate forecast. What is more, the continuity prior is shown to be well suited for low number of samples settings.

## 1 Introduction

Multivariate time series are ubiquitous in various domains of science, such as healthcare (Jensen et al., 2014), astronomy (Scargle, 1982), or climate science (Schneider, 2001). Much of the methodology for time-series analysis assumes that signals are measured systematically at fixed time intervals. However, much real-world data can be *sporadic* (*i.e.*, the signals are sampled irregularly and not all signals are measured each time). A typical example is patient measurements, which are taken when the patient comes for a visit (*e.g.,* sometimes skipping an appointment) and where not every measurement is taken at every visit. Modeling then becomes challenging as such data violates the main assumptions underlying traditional machine learning methods (such as recurrent neural networks).

Recently, the Neural Ordinary Differential Equation (ODE) model (Chen et al., 2018) opened the way for a novel, continuous representation of neural networks. As time is intrinsically continuous, this framework is particularly attractive for time-series analysis. It opens the perspective of tackling

---

[*]Both authors contributed equally
[†]Corresponding author

the issue of irregular sampling in a natural fashion, by integrating the dynamics over whatever time interval needed. Up to now however, such ODE dynamics have been limited to the continuous *generation* of observations (*e.g.*, decoders in variational auto-encoders (VAEs) (Kingma & Welling, 2013) or normalizing flows (Rezende et al., 2014)).

Instead of the encoder-decoder architecture where the ODE part is decoupled from the input processing, we introduce a tight integration by *interleaving* the ODE and the input processing steps. Conceptually, this allows us to drive the dynamics of the ODE directly by the incoming sporadic inputs. To this end, we propose (1) a continuous time version of the Gated Recurrent Unit and (2) a Bayesian update network that processes the sporadic observations. We combine these two ideas to form the GRU-ODE-Bayes method.

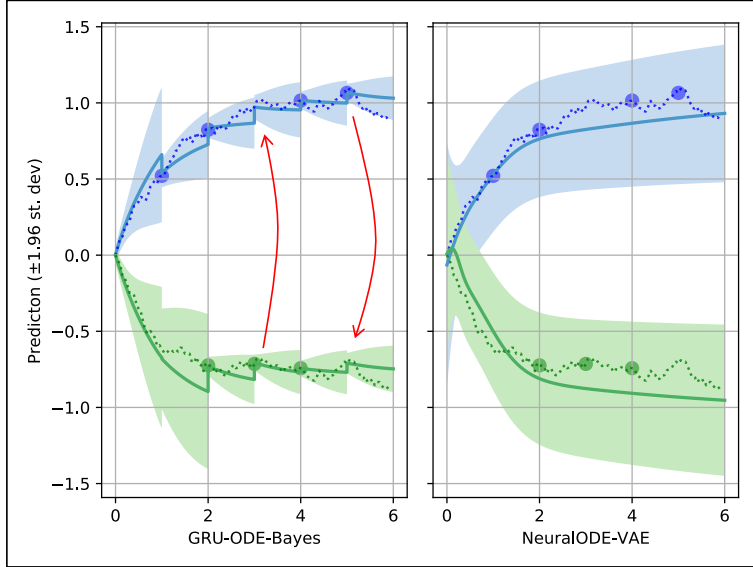

Figure 1: Comparison of GRU-ODE-Bayes and NeuralODE-VAE on a 2D Ornstein-Uhlenbeck process with highly correlated Wiener processes ($\rho = 0.99$). Dots are the values of the actual underlying process (dotted lines) from which the sporadic observations are obtained. Solid lines and shaded areas are the inferred means and 95% confidence intervals. Note the smaller errors and smaller variance of GRU-ODE-Bayes vs. NeuralODE-VAE. Note also that GRU-ODE-Bayes can infer that a jump in one variable also implies a jump in the other unobserved one (red arrows). Similarly, it also learns the reduction of variance resulting from a new incoming observation.

The tight coupling between observation processing and ODE dynamics allows the proposed method to model fine-grained nonlinear dynamical interactions between the variables. As illustrated in Figure 1, GRU-ODE-Bayes can (1) quickly infer the unknown parameters of the underlying stochastic process and (2) learn the correlation between its variables (red arrows in Figure 1). In contrast, the encoder-decoder based method NeuralODE-VAE proposed by Chen et al. (2018) captures the general structure of the process without being able to recover detailed interactions between the variables (see Section 4 for detailed comparison).

Our model enjoys important theoretical properties. We frame our analysis in a general way by considering that observations follow the dynamics driven by a stochastic differential equation (SDE). In Section 4 and Appendix I, we show that GRU-ODE-Bayes can exactly represent the corresponding Fokker-Planck dynamics in the special case of the Ornstein-Uhlenbeck process, as well as in generalized versions of it. We further perform an empirical evaluation and show that our method outperforms the state of the art on healthcare and climate data (Section 5).

## 1.1 Problem statement

We consider the general problem of forecasting on $N$ sporadically observed $D$-dimensional time series. For example, data from $N$ patients where $D$ clinical longitudinal variables can potentially be measured. Each time series $i \in \{1, \ldots, N\}$ is measured at $K_i$ time points specified by a vector

of observation times $\mathbf{t}_i \in \mathbb{R}^{K_i}$. The values of these observations are specified by a matrix of observations $\mathbf{y}_i \in \mathbb{R}^{K_i \times D}$ and an observation mask $\mathbf{m}_i \in \{0,1\}^{K_i \times D}$ (to indicate which of the variables are measured at each time point).

We assume that observations $\mathbf{y}_i$ are sampled from the realizations of a $D$-dimensional stochastic process $\mathbf{Y}(t)$ whose dynamics is driven by an unknown SDE:

$$d\mathbf{Y}(t) = \mu(\mathbf{Y}(t))dt + \sigma(\mathbf{Y}(t))d\mathbf{W}(t), \qquad (1)$$

where $d\mathbf{W}(t)$ is a Wiener process. The distribution of $\mathbf{Y}(t)$ then evolves according to the celebrated Fokker-Planck equation (Risken, 1996). We refer to the mean and covariance parameters of its probability density function (PDF) as $\mu_\mathbf{Y}(t)$ and $\Sigma_\mathbf{Y}(t)$.

Our goal will be to model the unknown temporal functions $\mu_\mathbf{Y}(t)$ and $\Sigma_\mathbf{Y}(t)$ from the sporadic measurements $\mathbf{y}_i$. These are obtained by sampling the random vectors $\mathbf{Y}(t)$ at times $\mathbf{t}_i$ with some observation noise $\boldsymbol{\epsilon}$. Not all dimensions are sampled each time, resulting in missing values in $\mathbf{y}_i$. In contrast to classical SDE inference (Särkkä & Solin, 2019), we consider the functions $\mu_\mathbf{Y}(t)$ and $\Sigma_\mathbf{Y}(t)$ are parametrized by neural networks.

This SDE formulation is general. It embodies the natural assumption that seemingly identical processes can evolve differently because of unobserved information. In the case of intensive care, as developed in Section 5, it reflects the evolving uncertainty regarding the patient's future condition.

## 2  Proposed method

At a high level, we propose a dual mode system consisting of (1) a GRU-inspired continuous-time state evolution (GRU-ODE) that *propagates* in time the hidden state $\mathbf{h}$ of the system between observations and (2) a network that *updates* the current hidden state to incorporate the incoming observations (GRU-Bayes). The system switches from propagation to update and back whenever a new observation becomes available.

We also introduce an observation model $f_{\text{obs}}(\mathbf{h}(t))$ mapping $\mathbf{h}$ to the estimated parameters of the observations distribution $\mu_\mathbf{Y}(t)$ and $\Sigma_\mathbf{Y}(t)$ (details in Appendix E). GRU-ODE then explicitly learns the Fokker-Planck dynamics of Eq. 1. This procedure allows end-to-end training of the system to minimize the loss with respect to the sporadically sampled observations $\mathbf{y}$.

### 2.1  GRU-ODE derivation

To derive the GRU-based ODE, we first show that the GRU proposed by Cho et al. (2014) can be written as a difference equation. First, let $\mathbf{r}_t$, $\mathbf{z}_t$, and $\mathbf{g}_t$ be the reset gate, update gate, and update vector of the GRU:

$$\begin{aligned} \mathbf{r}_t &= \sigma(W_r \mathbf{x}_t + U_r \mathbf{h}_{t-1} + \mathbf{b}_r) \\ \mathbf{z}_t &= \sigma(W_z \mathbf{x}_t + U_z \mathbf{h}_{t-1} + \mathbf{b}_z) \\ \mathbf{g}_t &= \tanh(W_h \mathbf{x}_t + U_h(\mathbf{r}_t \odot \mathbf{h}_{t-1}) + \mathbf{b}_h), \end{aligned} \qquad (2)$$

where $\odot$ is the elementwise product. Then the standard update for the hidden state $\mathbf{h}$ of the GRU is

$$\mathbf{h}_t = \mathbf{z}_t \odot \mathbf{h}_{t-1} + (1 - \mathbf{z}_t) \odot \mathbf{g}_t.$$

We can also write this as $\mathbf{h}_t = \mathbf{GRU}(\mathbf{h}_{t-1}, \mathbf{x}_t)$. By subtracting $\mathbf{h}_{t-1}$ from this state update equation and factoring out $(1 - \mathbf{z}_t)$, we obtain a difference equation

$$\begin{aligned} \Delta \mathbf{h}_t = \mathbf{h}_t - \mathbf{h}_{t-1} &= \mathbf{z}_t \odot \mathbf{h}_{t-1} + (1 - \mathbf{z}_t) \odot \mathbf{g}_t - \mathbf{h}_{t-1} \\ &= (1 - \mathbf{z}_t) \odot (\mathbf{g}_t - \mathbf{h}_{t-1}). \end{aligned}$$

This difference equation naturally leads to the following ODE for $\mathbf{h}(t)$:

$$\frac{d\mathbf{h}(t)}{dt} = (1 - \mathbf{z}(t)) \odot (\mathbf{g}(t) - \mathbf{h}(t)), \qquad (3)$$

where $\mathbf{z}$, $\mathbf{g}$, $\mathbf{r}$ and $\mathbf{x}$ are the continuous counterpart of Eq. 2. See Appendix A for the explicit form.

We name the resulting system *GRU-ODE*. Similarly, we derive the *minimal GRU-ODE*, a variant based on the minimal GRU (Zhou et al., 2016), described in appendix G.

In case *continuous observations* or *control signals* are available, they can be naturally fed to the GRU-ODE input $\mathbf{x}(t)$. For example, in the case of clinical trials, the administered daily doses of the drug under study can be used to define a continuous input signal. If no continuous input is available, then nothing is fed as $\mathbf{x}(t)$ and the resulting ODE in Eq. 3 is autonomous, with $\mathbf{g}(t)$ and $\mathbf{z}(t)$ only depending on $\mathbf{h}(t)$.

## 2.2 General properties of GRU-ODE

GRU-ODE enjoys several useful properties:

**Boundedness.** First, the hidden state $\mathbf{h}(t)$ stays within the $[-1, 1]$ range[3]. This restriction is crucial for the compatibility with the GRU-Bayes model and comes from the negative feedback term in Eq. 3, which stabilizes the resulting system. In detail, if the $j$-th dimension of the starting state $\mathbf{h}(0)$ is within $[-1, 1]$, then $\mathbf{h}(t)_j$ will always stay within $[-1, 1]$ because

$$\left. \frac{d\mathbf{h}(t)_j}{dt} \right|_{t:\mathbf{h}(t)_j=1} \leq 0 \quad \text{and} \quad \left. \frac{d\mathbf{h}(t)_j}{dt} \right|_{t:\mathbf{h}(t)_j=-1} \geq 0.$$

This can be derived from the ranges of $\mathbf{z}$ and $\mathbf{g}$ in Eq. 2. Moreover, would $\mathbf{h}(0)$ start outside of the $[-1, 1]$ region, the negative feedback would quickly push $\mathbf{h}(t)$ into this region, making the system also robust to numerical errors.

**Continuity.** Second, GRU-ODE is Lipschitz continuous with constant $K = 2$. Importantly, this means that GRU-ODE encodes a *continuity prior* for the latent process $\mathbf{h}(t)$. This is in line with the assumption of a continuous hidden process generating observations (Eq. 1). In Section 5.5, we demonstrate empirically the importance of this prior in the small-sample regime.

**General numerical integration.** As a parametrized ODE, GRU-ODE can be integrated with any numerical solver. In particular, adaptive step size solvers can be used. Our model can then afford large time steps when the internal dynamics is slow, taking advantage of the continuous time formulation of Eq. 3. It can also be made faster with sophisticated ODE integration methods. We implemented the following methods: Euler, explicit midpoint, and Dormand-Prince (an adaptive step size method). Appendix C illustrates that the Dormand-Prince method requires fewer time steps.

## 2.3 GRU-Bayes

GRU-Bayes is the module that processes the sporadically incoming observations to update the hidden vectors, and hence the estimated PDF of $\mathbf{Y}(t)$. This module is based on a standard GRU and thus operates in the region $[-1, 1]$ that is required by GRU-ODE. In particular, GRU-Bayes is able to update $\mathbf{h}(t)$ to any point in this region. Any adaptation is then within reach with a *single observation*.

To feed the GRU unit inside GRU-Bayes with a non-fully-observed vector, we first preprocess it with an observation mask using $f_{\text{prep}}$, as described in Appendix D. For a given time series, the resulting update for its $k$-th observation $\mathbf{y}[k]$ at time $t = \mathbf{t}[k]$ with mask $\mathbf{m}[k]$ and hidden vector $\mathbf{h}(t_-)$ is

$$\mathbf{h}(t_+) = \mathbf{GRU}(\mathbf{h}(t_-), f_{\text{prep}}(\mathbf{y}[k], \mathbf{m}[k], \mathbf{h}(t_-))), \tag{4}$$

where $\mathbf{h}(t_-)$ and $\mathbf{h}(t_+)$ denote the hidden representation before and after the jump from GRU-Bayes update. We also investigate an alternative option where the $\mathbf{h}(t)$ is updated by each observed dimension *sequentially*. We call this variant *GRU-ODE-Bayes-seq* (see Appendix F for more details). In Appendix H, we run an ablation study of the proposed GRU-Bayes architecture by replacing it with a MLP and show that the aforementioned properties are crucial for good performance.

## 2.4 GRU-ODE-Bayes

The proposed GRU-ODE-Bayes combines GRU-ODE and GRU-Bayes. The GRU-ODE is used to evolve the hidden state $\mathbf{h}(t)$ in continuous time between the observations and GRU-Bayes transforms the hidden state, based on the observation $\mathbf{y}$, from $\mathbf{h}(t_-)$ to $\mathbf{h}(t_+)$. As best illustrated in Figure 2, the alternation between GRU-ODE and GRU-Bayes results in an ODE with *jumps*, where the jumps are at the locations of the observations.

GRU-ODE-Bayes is best understood as a filtering approach. Based on previous observations (until time $t_k$), it can estimate the probability of future observations. Like the celebrated Kalman filter, it alternates between a *prediction* (GRU-ODE) and a *filtering* (GRU-Bayes) phase. Future values of the time series are predicted by integrating the hidden process $\mathbf{h}(t)$ in time, as shown on the green solid line in Figure 2. The update step discretely updates the hidden state when a new measurement becomes available (dotted blue line). Let's note that unlike the Kalman filter, our approach is able to learn complex dynamics for the hidden process.

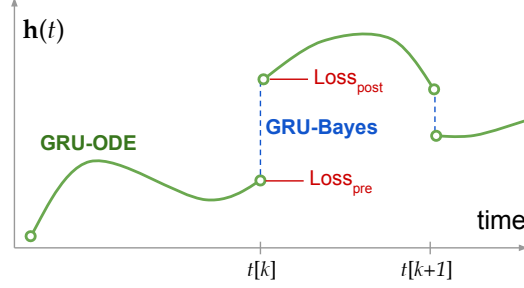

Figure 2: GRU-ODE-Bayes uses GRU-ODE to evolve the hidden state between two observation times $t[k]$ and $t[k + 1]$. GRU-Bayes processes the observations and updates the hidden vector $\mathbf{h}$ in a discrete fashion, reflecting the additional information brought in by the observed data.

**Objective function**

To train the model using sporadically-observed samples, we introduce two losses. The first loss, $\mathrm{Loss}_{\mathrm{pre}}$, is computed before the observation update and is the negative log-likelihood (NegLL) of the observations. For the observation of a single sample, we have (for readability we drop the time indexing):

$$\mathrm{Loss}_{\mathrm{pre}} = -\sum_{j=1}^{D} m_j \log p(y_j|\theta = f_{\mathrm{obs}}(\mathbf{h}_-)_j),$$

where $m_j$ is the observation mask and $f_{\mathrm{obs}}(\mathbf{h}_-)_j$ are the parameters of the distribution before the update, for dimension $j$. Thus, the error is only computed on the observed values of $\mathbf{y}$.

For the second loss, let $p_{\mathrm{pre}}$ denote the predicted distributions (from $\mathbf{h}_-$) before GRU-Bayes.

With $p_{\mathrm{obs}}$, the PDF of $\mathbf{Y}(t)$ given the noisy observation (with noise vector $\boldsymbol{\epsilon}$), we first compute the analogue of the Bayesian update:

$$p_{\mathrm{Bayes},j} \propto p_{\mathrm{pre},j} \cdot p_{\mathrm{obs},j}.$$

Let $p_{\mathrm{post}}$ denote the predicted distribution (from $\mathbf{h}_+$) after applying GRU-Bayes. We then define the post-jump loss as the KL-divergence between $p_{\mathrm{Bayes}}$ and $p_{\mathrm{post}}$:

$$\mathrm{Loss}_{\mathrm{post}} = \sum_{j=1}^{D} m_j D_{KL}(p_{\mathrm{Bayes},j}||p_{\mathrm{post},j}).$$

In this way, we force our model to learn to mimic a Bayesian update.

Similarly to the pre-jump loss, $\mathrm{Loss}_{\mathrm{post}}$ is computed only for the observed dimensions. The total loss is then obtained by adding both losses with a weighting parameter $\lambda$.

---

**Algorithm 1** GRU-ODE-Bayes

**Input:** Initial state $\mathbf{h}_0$,
      observations $\mathbf{y}$, mask $\mathbf{m}$,
      observation times $\mathbf{t}$, final time $T$.
Initialize time $= 0, \mathrm{loss} = 0, \mathbf{h} = \mathbf{h}_0$.
**for** $k = 1$ **to** $K$ **do**
    {ODE evolution to $\mathbf{t}[k]$}
    $\mathbf{h} = \mathbf{GRU\text{-}ODE}(\mathbf{h}, \mathrm{time}, \mathbf{t}[k])$
    $\mathrm{time} = t[k]$
    {Pre-jump loss}
    $\mathrm{loss} \mathrel{+}= \mathrm{Loss}_{\mathrm{pre}}(\mathbf{y}[k], \mathbf{m}[k], \mathbf{h})$
    {Update}
    $\mathbf{h} = \mathbf{GRU\text{-}Bayes}(\mathbf{y}[k], \mathbf{m}[k], \mathbf{h})$
    {Post-jump loss}
    $\mathrm{loss} \mathrel{+}= \lambda . \mathrm{Loss}_{\mathrm{post}}(\mathbf{y}[k], \mathbf{m}[k], \mathbf{h})$
**end for**
{ODE evolution to $T$}
$\mathbf{h} = \mathbf{GRU\text{-}ODE}(\mathbf{h}, \mathbf{t}[N_K], T)$
**return** $(\mathbf{h}, \mathrm{loss})$

---

For binomial and Gaussian distributions, computing $\text{Loss}_{\text{post}}$ can be done analytically. In the case of Gaussian distribution we can compute the Bayesian updated mean $\mu_{\text{Bayes}}$ and variance $\sigma_{\text{Bayes}}^2$ as

$$\mu_{\text{Bayes}} = \frac{\sigma_{\text{obs}}^2}{\sigma_{\text{pre}}^2 + \sigma_{\text{obs}}^2} \mu_{\text{pre}} + \frac{\sigma_{\text{pre}}^2}{\sigma_{\text{pre}}^2 + \sigma_{\text{obs}}^2} \mu_{\text{obs}}$$

$$\sigma_{\text{Bayes}}^2 = \frac{\sigma_{\text{pre}}^2 \cdot \sigma_{\text{obs}}^2}{\sigma_{\text{pre}}^2 + \sigma_{\text{obs}}^2},$$

where for readability we dropped the dimension sub-index. In many real-world cases, the observation noise $\sigma_{\text{obs}}^2 \ll \sigma_{\text{pre}}^2$, in which case $p_{\text{Bayes}}$ is just the observation distribution: $\mu_{\text{Bayes}} = \mu_{\text{obs}}$ and $\sigma_{\text{Bayes}}^2 = \sigma_{\text{obs}}^2$.

## 2.5 Implementation

The pseudocode of GRU-ODE-Bayes is depicted in Algorithm 1, where a forward pass is shown for a single time series [4]. For mini-batching several time series we sort the observation times across all time series and for each unique time point $\mathbf{t}[k]$, we create a list of the time series that have observations. The main loop of the algorithm iterates over this set of unique time points. In the GRU-ODE step, we propagate all hidden states jointly. The GRU-Bayes update and the loss calculation are only executed on the time series that have observation at that particular time point. The complexity of our approach then scales linearly with the number of observations and quadratically with the dimension of the observations. When memory cost is a bottleneck, the gradient can be computed using the adjoint method, without backpropagating through the solver operations (Chen et al., 2018).

# 3 Related research

Machine learning has a long history in time series modelling (Mitchell, 1999; Gers et al., 2000; Wang et al., 2006; Chung et al., 2014). However, recent massive real-world data collection, such as electronic health records (EHR), increase the need for models capable of handling such complex data (Lee et al., 2017). As stated in the introduction, their *sporadic* nature is the main difficulty.

To address the nonconstant sampling, a popular approach is to recast observations into fixed duration time bins. However, this representation results in missing observation both in time and across features dimensions. This makes the direct usage of neural network architectures tricky. To overcome this issue, the main approach consists in some form of data imputation and jointly feeding the observation mask and times of observations to the recurrent network (Che et al., 2018; Choi et al., 2016a; Lipton et al., 2016; Du et al., 2016; Choi et al., 2016b; Cao et al., 2018). This approach strongly relies on the assumption that the network will learn to process true and imputed samples differently. Despite some promising experimental results, there is no guarantee that it will do so. Some researchers have tried to alleviate this limitation by introducing more meaningful data representation for sporadic time series (Rajkomar et al., 2018; Razavian & Sontag, 2015; Ghassemi et al., 2015), like tensors (De Brouwer et al., 2018; Simm et al., 2017).

Others have addressed the missing data problem with generative probabilistic models. Among those, (multitask) Gaussian processes (GP) are the most popular by far (Bonilla et al., 2008). They have been used for smart imputation before a RNN or CNN architecture (Futoma et al., 2017; Moor et al., 2019), for modelling a hidden process in joint models (Soleimani et al., 2018), or to derive informative representations of time series (Ghassemi et al., 2015). GPs have also been used for direct forecasting (Cheng et al., 2017). However, they usually suffer from high uncertainty outside the observation support, are computationally intensive (Quiñonero-Candela & Rasmussen, 2005), and learning the optimal kernel is tricky. Neural Processes, a neural version of GPs, have also been introduced by Garnelo et al. (2018). In contrast with our work that focuses on continuous-time *real-valued* time series, continuous time modelling of time-to-events has been addressed with point processes (Mei & Eisner, 2017) and continuous time Bayesian networks (Nodelman et al., 2002). Yet, our continuous modelling of the latent process allows us to straightforwardly model a continuous intensity function and thus handle both real-valued and event type of data. This extension was left for future work.

Most recently, the seminal work of Chen et al. (2018) suggested a continuous version of neural networks that overcomes the limits imposed by discrete-time recurrent neural networks. Coupled with a variational auto-encoder architecture (Kingma & Welling, 2013), it proposed a natural way of generating irregularly sampled data. However, it transferred the difficult task of processing sporadic data to the encoder, which is a discrete-time RNN. In a work submitted concomitantly to ours (Rubanova et al., 2019), the authors proposed a convincing new VAE architecture that uses a Neural-ODE architecture for both encoding and decoding the data.

Related auto-encoder approaches with sequential latents operating in discrete time have also been proposed (Krishnan et al., 2015, 2017). These models rely on classical RNN architectures in their inference networks, hence not addressing the sporadic nature of the data. What is more, if they have been shown useful for smoothing and counterfactual inference, their formulation is less suited for forecasting. Our method also has connections to the Extended Kalman Filter (EKF) that models the dynamics of the distribution of processes in continuous time. However, the practical applicability of the EKF is limited because of the linearization of the state update and the difficulties involved in identifying its parameters. Importantly, the ability of the GRU to learn long-term dependencies is a significant advantage.

Finally, other works have investigated the relationship between deep neural networks and partial differential equations. An interesting line of research has focused on deriving better deep architectures motivated by the stability of the corresponding patial differential equations (PDE) (Haber & Ruthotto, 2017; Chang et al., 2019). Despite their PDE motivation, those approaches eventually designed new *discrete* architectures and didn't explore the application on continuous inputs and time.

## 4 Application to synthetic SDEs

Figure 1 illustrates the capabilities of our approach compared to NeuralODE-VAE on data generated from a process driven by a multivariate Ornstein-Uhlenbeck (OU) SDE with random parameters. Compared to NeuralODE-VAE, which retrieves the average dynamics of the samples, our approach detects the correlation between both features and updates its predictions more finely as new observations arrive. In particular, note that GRU-ODE-Bayes updates its prediction and confidence on a feature even when only the other one is observed, taking advantage from the fact that they are correlated. This can be seen on the left pane of Figure 1 where at time $t = 3$, Dimension 1 (blue) is updated because of the observation of Dimension 2 (green).

By directly feeding sporadic inputs into the ODE, GRU-ODE-Bayes sequentially *filters* the hidden state and thus estimates the PDF of the future observations. This is the core strength of the proposed method, allowing it to perform long-term predictions.

In Appendix I, we further show that our model can exactly represent the dynamics of multivariate OU process with random variables. Our model can also handle nonlinear SDEs as shown in Appendix J where we present an example inspired by the Brusselator (Prigogine, 1982), a chaotic ODE.

## 5 Empirical evaluation

We evaluated our model on two data sets from different application areas: healthcare and climate forecasting. In both applications, we assume the data consists of noisy observations from an underlying unobserved latent process as in Eq. 1. We focused on the general task of forecasting the time series at future time points. Models are trained to minimize negative log-likelihood.

### 5.1 Baselines

We used a comprehensive set of state-of-the-art baselines to compare the performance of our method. All models use the same hidden size representation and comparable number of parameters.

**NeuralODE-VAE** (Chen et al., 2018). We model the time derivative of the hidden representation as a 2-layer MLP. To take missingness across features into account, we add a mechanism to feed an observation mask.

**Imputation Methods.** We implemented two imputation methods as described in Che et al. (2018): *GRU-Simple* and *GRU-D*.

**Sequential VAEs** (Krishnan et al., 2015, 2017). We extended the deep Kalman filter architecture by feeding an observation mask and updating the loss function accordingly.

**T-LSTM** (Baytas et al., 2017). We reused the proposed time-aware LSTM cell to design a forecasting RNN with observation mask.

## 5.2 Electronic health records

Electronic Health Records (EHR) analysis is crucial to achieve data-driven personalized medicine (Lee et al., 2017; Goldstein et al., 2017; Esteva et al., 2019). However, efficient modeling of this type of data remains challenging. Indeed, it consists of *sporadically* observed longitudinal data with the extra hurdle that there is no standard way to align patients trajectories (*e.g.*, at hospital admission, patients might be in very different state of progression of their condition). Those difficulties make EHR analysis well suited for GRU-ODE-Bayes.

We use the publicly available MIMIC-III clinical database (Johnson et al., 2016), which contains EHR for more than 60,000 critical care patients. We select a subset of 21,250 patients with sufficient observations and extract 96 different longitudinal real-valued measurements over a period of 48 hours after patient admission. We refer the reader to Appendix K for further details on the cohort selection. We focus on the predictions of the next 3 measurements after a 36-hour observation window.

## 5.3 Climate forecast

From short-term weather forecast to long-range prediction or assessment of systemic changes, such as global warming, climatic data has always been a popular application for time-series analysis. This data is often considered to be regularly sampled over long periods of time, which facilitates their statistical analysis. Yet, this assumption does not usually hold in practice. Missing data are a problem that is repeatedly encountered in climate research because of, among others, measurement errors, sensor failure, or faulty data acquisition. The actual data is then sporadic and researchers usually resort to imputation before statistical analysis (Junninen et al., 2004; Schneider, 2001).

We use the publicly available United State Historical Climatology Network (USHCN) daily data set (Menne et al.), which contains measurements of 5 climate variables (daily temperatures, precipitation, and snow) over 150 years for 1,218 meteorological stations scattered over the United States. We selected a subset of 1,114 stations and an observation window of 4 years (between 1996 and 2000). To make the time series sporadic, we subsample the data such that each station has an average of around 60 observations over those 4 years. Appendix L contains additional details regarding this procedure. The task is then to predict the next 3 measurements after the first 3 years of observation.

## 5.4 Results

We report the performance using 5-fold cross-validation. Hyperparameters (dropout and weight decay) are chosen using an inner holdout validation set (20%) and performance are assessed on a left-out test set (10%). Those folds are reused for each model we evaluated for sake of reproducibility and fair comparison (More details in Appendix O). Performance metrics for both tasks (NegLL and MSE) are reported in Table 1. GRU-ODE-Bayes handles the sporadic data more naturally and can more finely model the dynamics and correlations between the observed features, which results in higher performance than other methods for both data sets. In particular, GRU-ODE-Bayes unequivocally outperforms all other methods on both data sets.

## 5.5 Impact of continuity prior

To illustrate the capabilities of the derived GRU-ODE cell presented in Section 2.1, we consider the case of time series forecasting with low sample size. In the realm of EHR prediction, this could be framed as a *rare disease* setup, where data is available for few patients only. In this case of scarce number of samples, the continuity prior embedded in GRU-ODE is crucially important as it provides important prior information about the underlying process.

To highlight the importance of the GRU-ODE cell, we compare two versions of our model : the classical GRU-ODE-Bayes and one where the GRU-ODE cell is replaced by a discretized autonomous GRU. We call the latter *GRU-Discretized-Bayes*. Table 2 shows the results for MIMIC-III with varying

Table 1: Forecasting results.

| | USHCN-DAILY | | MIMIC-III | |
|---|---|---|---|---|
| MODEL | MSE | NEGLL | MSE | NEGLL |
| NEURALODE-VAE | $0.96 \pm 0.11$ | $1.46 \pm 0.10$ | $0.89 \pm 0.01$ | $1.35 \pm 0.01$ |
| NEURALODE-VAE-MASK | $0.83 \pm 0.10$ | $1.36 \pm 0.05$ | $0.89 \pm 0.01$ | $1.36 \pm 0.01$ |
| SEQUENTIAL VAE | $0.83 \pm 0.07$ | $1.37 \pm 0.06$ | $0.92 \pm 0.09$ | $1.39 \pm 0.07$ |
| GRU-SIMPLE | $0.75 \pm 0.12$ | $1.23 \pm 0.10$ | $0.82 \pm 0.05$ | $1.21 \pm 0.04$ |
| GRU-D | $0.53 \pm 0.06$ | $0.99 \pm 0.07$ | $0.79 \pm 0.06$ | $1.16 \pm 0.05$ |
| T-LSTM | $0.59 \pm 0.11$ | $1.67 \pm 0.50$ | $0.62 \pm 0.05$ | $1.02 \pm 0.02$ |
| GRU-ODE-BAYES | $\mathbf{0.43} \pm 0.07$ | $\mathbf{0.84} \pm 0.11$ | $\mathbf{0.48} \pm 0.01$ | $\mathbf{0.83} \pm 0.04$ |

number of patients in the training set. While our discretized version matches the continuous one on the full data set, GRU-ODE cell achieves higher accuracy when the number of samples is low, highlighting the importance of the continuity prior. Log-likelihood results are given in Appendix M.

Table 2: Comparison between GRU-ODE and discretized version in the small-sample regime (MSE).

| MODEL | 1,000 PATIENTS | 2,000 PATIENTS | FULL |
|---|---|---|---|
| NEURALODE-VAE-MASK | $0.94 \pm 0.01$ | $0.94 \pm 0.01$ | $0.89 \pm 0.01$ |
| GRU-DISCRETIZED-BAYES | $0.87 \pm 0.02$ | $0.77 \pm 0.02$ | $\mathbf{0.46} \pm 0.05$ |
| GRU-ODE-BAYES | $\mathbf{0.77} \pm 0.01$ | $\mathbf{0.72} \pm 0.01$ | $\mathbf{0.48}^{\dagger} \pm 0.01$ |

# 6 Conclusion and future work

We proposed a model combining two novel techniques, GRU-ODE and GRU-Bayes, which allows feeding sporadic observations into a continuous ODE dynamics describing the evolution of the probability distribution of the data. Additionally, we showed that this filtering approach enjoys attractive representation capabilities. Finally, we demonstrated the value of GRU-ODE-Bayes on both synthetic and real-world data. Moreover, while a discretized version of our model performed well on the full MIMIC-III data set, the continuity prior of our ODE formulation proves particularly important in the small-sample regime, which is particularly relevant for real-world clinical data where many data sets remain relatively modest in size.

In this work, we focused on time-series data with Gaussian observations. However, GRU-ODE-Bayes can also be extended to binomial and multinomial observations since the respective NegLL and KL-divergence are analytically tractable. This allows the modeling of sporadic observations of both discrete and continuous variables.

## Acknowledgements

Edward De Brouwer is funded by a FWO-SB grant. Yves Moreau is funded by (1) Research Council KU Leuven: C14/18/092 SymBioSys3; CELSA-HIDUCTION, (2) Innovative Medicines Initiative: MELLODY, (3) Flemish Government (ELIXIR Belgium, IWT, FWO 06260) and (4) Impulsfonds AI: VR 2019 2203 DOC.0318/1QUATER Kenniscentrum Data en Maatschappij. Computational resources and services used in this work were provided by the VSC (Flemish Supercomputer Center), funded by the Research Foundation - Flanders (FWO) and the Flemish Government – department EWI. We also gratefully acknowledge the support of NVIDIA Corporation with the donation of the Titan Xp GPU used for this research.

---

[†]Statistically not different from best (p-value $> 0.6$ with paired t-test).

## Footnotes

[3]We use the notation $[-1, 1]$ to also mean multi-dimensional range (*i.e.*, all elements are within $[-1, 1]$).

[4] Code is available in the following anonymous repository : `https://github.com/edebrouwer/gru_ode_bayes`

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
