[Supplementary Material]

# A    Full formulation of the GRU-ODE cell

The full ODE equation for GRU-ODE is the following:

$$\frac{d\mathbf{h}(t)}{dt} = (1 - \mathbf{z}(t)) \odot (\mathbf{g}(t) - \mathbf{h}(t)),$$

with

$$\begin{aligned}
\mathbf{r}(t) &= \sigma(W_r\mathbf{x}(t) + U_r\mathbf{h}(t) + \mathbf{b}_r) \\
\mathbf{z}(t) &= \sigma(W_z\mathbf{x}(t) + U_z\mathbf{h}(t) + \mathbf{b}_z) \\
\mathbf{g}(t) &= \tanh(W_h\mathbf{x}(t) + U_h(\mathbf{r}(t) \odot \mathbf{h}(t)) + \mathbf{b}_h).
\end{aligned}$$

Matrices $W \in \mathbb{R}^{H \times D}$, $B \in \mathbb{R}^{H \times H}$ and bias vectors $\mathbf{b} \in \mathbb{R}^H$ are the parameters of the cell. $H$ and $D$ are the dimension of the hidden process and of the inputs respectively.

# B    Lipschitz continuity of GRU-ODE

As $\mathbf{h}$ is differentiable and continous on $t$, we know from the mean value theorem that for any $t_a, t_b \in t$, there exists $t^* \in (t_a, t_b)$ such that

$$\mathbf{h}(t_b) - \mathbf{h}(t_a) = \frac{d\mathbf{h}}{dt} \mid_{t^*} (t_b - t_a).$$

Taking the euclidean norm of the previous expression, we find

$$\mid \mathbf{h}(t_b) - \mathbf{h}(t_a) \mid = \mid \frac{d\mathbf{h}}{dt} \mid_{t^*} (t_b - t_a) \mid .$$

Furthermore, we showed that $\mathbf{h}$ is bounded on $[-1, 1]$. Hence, because of the bounded functions appearing in the ODE (sigmoids and hyperbolic tangents), the derivative of $\mathbf{h}$ is itself bounded by $[-2, 2]$. We conclude that $\mathbf{h}(t)$ is Lipschitz continuous with constant $K = 2$.

# C    Comparison of numerical integration methods

We implemented three numerical integration methods, among which the classical Euler method and the Dormand-Prince method (DOPRI). DOPRI is a popular adaptive step size numerical integration method relying on 2 Runge-Kutta solvers of order 4 and 5. The advantage of adaptive step size methods is that they can tune automatically the number of steps to integrate the ODE until the desired point.

Figure 3 illustrates the number of steps taken by both solvers when given the same data and same ODE. We observe that using an adaptive step size results in half as many time steps. More steps are taken near the observations and as the underlying process becomes smoother, the step size increase, as observed on the right side of the figure. However, each time step requires significantly fewer computations for Euler than for DOPRI, so that Euler's method appears more than competitive on the data and simulations we have considered so far. Nevertheless, DOPRI might still be preferred as default method because of its better numerical stability.

Figure 3: Comparison of Euler and DOPRI numerical integration methods for same inputs and same ODE. Colored ticks on the $x$ axis represent the evaluation time for each method. Dotted lines show the evolution of the estimated mean distribution of the observations while the dots stand for the observations fed to the model.

## D  Mapping to deal with missingness across features

The preprocessing step $f_{\mathrm{prep}}$ for GRU-Bayes takes in the hidden state $\mathbf{h}$ and computes the parameters for the observation PDFs $\theta = f_{\mathrm{obs}}(\mathbf{h}(t))$. In the case of a Gaussian, $\theta_d$ contains the means and log-variances for dimension $d$ of $\mathbf{Y}(t)$. Then, we create a vector $\mathbf{q}_d$ that concatenates $\theta_d$ with the observed value $\mathbf{y}[k]_d$ and the normalized error term, which for the Gaussian case is $(\mathbf{y}[k]_d - \mu_d)/\sigma_d$, where $\mu_d$ and $\sigma_d$ are the mean and standard deviation derived from $\theta_d$. We then multiply the vectors $\mathbf{q}_d$ by a dimension-specific weight matrix $\mathbf{W}_d$ and apply a ReLU nonlinearity. Next, we zero all results that did not have an observation (by multiplying them with mask $m_d$). Finally, the concatenation of the results is fed into the GRU unit of GRU-Bayes.

## E  Observation model mapping

The mapping from hidden $\mathbf{h}$ to the parameters of the distribution $\mu_{\mathbf{Y}(t)}$ and $\log(\Sigma_{\mathbf{Y}(t)})$. For this purpose we use a classical multi-layer perceptron architecture with a 25 dimensional hidden layer. Note that me map to the log of the variance in order to keep it positive.

## F  GRU-ODE-Bayes-seq

On top of the architecture described in the main bulk of this paper, we also propose a variant which process the sporadic inputs *sequentially*. In other words, GRU-Bayes will update its prediction on the hidden $\mathbf{h}$ for one input dimension after the other rather than jointly. We call this approach GRU-ODE-Bayes-seq.

In this sequential approach for GRU-Bayes, we process one-by-one all dimensions of $\mathbf{y}[k]$ that were *observed* at time $t[k]$ by first applying the preprocessing to each and then sending them to the GRU unit. The preprocessing steps are the same as in the nonsequential scheme (Appendix D) but without concatenation at the end because only one dimension is processed at a time. Note that the preprocessing of dimensions cannot be done in parallel as the hidden state $\mathbf{h}$ changes after each dimension is processed, which affects the computed $\theta_d$ and thus the resulting vector $\mathbf{q}_d$.

# G Minimal GRU-ODE

Following the same reasoning as for the full GRU cell, we also derived the minimal GRU-ODE cell, based on the minimal GRU cell. The minimal GRU writes :

$$\mathbf{f}_t = \sigma(W_f \mathbf{x}_t + U_f \mathbf{h}_{t-1} + \mathbf{b}_f)$$
$$\mathbf{h}_t = \mathbf{f}_t \odot \mathbf{h}_{t-1} + (1 - \mathbf{f}_t) \odot \sigma(W_h \mathbf{x}_t + U_h(\mathbf{h}_{t-1} \odot \mathbf{f}_t) + \mathbf{b}_h)$$

This can be rewritten as the following difference equation :

$$\Delta \mathbf{h}_t = (1 - \mathbf{f}_t) \odot (\sigma(W_h \mathbf{x}_t + U_h(\mathbf{h}_{t-1} \odot \mathbf{f}_t) + \mathbf{b}_h) - \mathbf{h}_{t-1})$$

Which leads to the corresponding ODE :

$$\frac{d\mathbf{h}(t)}{dt} = (1 - \mathbf{f}(t)) \odot (\sigma(W_h \mathbf{x}(t) + U_h(\mathbf{h}(t) \odot \mathbf{f}(t)) + \mathbf{b}_h) - \mathbf{h}(t))$$

# H Ablation study of GRU-Bayes

In order to demonstrate the fitness of the GRU-Bayes module for our architecture, we ran an ablation study where we replaced the GRU-Bayes with a 2 layers multi-layer perceptron. We used a *tanh* activation function for the hidden units and a linear activation for the output layer. We evaluate the performance of this modified architecture on the MIMIC dataset for the forecasting task. The results are presented in table 3. The proposed architecture outperforms a simple MLP module, due to the properties described in sections 2.2 and 2.3.

Table 3: NegLL and MSE results for proposed GRU-Bayes module and replaced with MLP.

| MODEL | MSE | NEGLL |
|---|---|---|
| GRU-BAYES ORIGINAL | $0.48 \pm 0.01$ | $0.83 \pm 0.04$ |
| GRU-BAYES MLP | $0.54 \pm 0.05$ | $1.05 \pm 0.02$ |

# I Application to Ornstein-Uhlenbeck SDEs

We demonstrate the capabilities of our approach on data generated from a process driven by an SDE as in Eq. 1. In particular, we focus on extensions of the multidimensional correlated Ornstein-Uhlenbeck (OU) process with varying parameters. For a particular sample $i$, the dynamics is given by the following SDE:

$$d\mathbf{Y}_i(t) = \theta_i(\mathbf{r}_i - \mathbf{Y}_i(t))dt + \sigma_i d\mathbf{W}(t), \tag{5}$$

where $\mathbf{W}(t)$ is a $D$-dimensional correlated Wiener process, $\mathbf{r}_i$ is the vector of targets, and $\theta_i$ is the reverting strength constant. For simplicity, we consider $\theta_i$ and $\sigma_i$ parameters as scalars. Each sample $\mathbf{y}_i$ is then obtained via the realization of process (5) with sample-specific parameters.

## I.1 Representation capabilities

We now show that our model exactly captures the dynamics of the distribution of $\mathbf{Y}(t)$ as defined in Eq. 5. The evolution of the PDF of a diffusion process is given by the corresponding Fokker-Planck equation. For the OU process, this PDF is Gaussian with time-dependent mean and covariance. Conditioned on a previous observation at time $t^*$, this gives

$$\mathbf{Y}_i(t) \mid \mathbf{Y}_i(t^*) \sim \mathcal{N}(\mu_{\mathbf{Y}}(t, t^*), \sigma^2_{\mathbf{Y}}(t, t^*)),$$
$$\mu_{\mathbf{Y}}(t, t^*) = \mathbf{r}_i + (\mathbf{Y}_i(t^*) - \mathbf{r}_i) \exp(-\theta_i(t - t^*)),$$
$$\sigma^2_{\mathbf{Y}}(t, t^*) = \frac{\sigma_i^2}{2\theta_i}(1 - \exp(-2\theta_i(t - t^*))).$$

Correlation of $\mathbf{Y}(t)$ is constant and equal to $\rho$, the correlation of the Wiener processes. The dynamics of the mean and variance parameters can be better expressed in the following ODE form:

$$\frac{d\mu_{\mathbf{Y}}(t,t^*)}{dt} = -\theta_i(\mu_{\mathbf{Y}}(t,t^*) - \mathbf{r}_i)$$
$$\frac{d\sigma_{\mathbf{Y}}^2(t,t^*)}{dt} = -2\theta_i\left(\sigma_{\mathbf{Y}}^2(t,t^*) - \frac{\sigma_i^2}{2\theta_i}\right) \tag{6}$$

With initial conditions $\mu_{\mathbf{Y}}(0,t^*) = \mathbf{Y}(t^*)$ and $\sigma_{\mathbf{Y}}^2(0,t^*) = 0$. We next investigate how specific versions of this ODE can be represented by our GRU-ODE-Bayes.

### I.1.1 Standard Ornstein-Uhlenbeck process

In standard OU, the parameters $\mathbf{r}_i$, $\sigma_i$, and $\theta_i$ are fixed and identical for all samples. The ODE (6) is linear and can then be represented directly with GRU-ODE by storing $\mu_{\mathbf{Y}}(t,t^*)$ and $\sigma_{\mathbf{Y}}^2(t,t^*)$ in the hidden state $\mathbf{h}(t)$ and matching the Equations (3) and (6). The OU parameters $\mathbf{r}_i$, $\sigma_i$ and $\theta_i$ are learned and encoded in the weights of GRU-ODE. GRU-Bayes then updates the hidden state and stores $\mu_{\mathbf{Y}}(t,t^*)$ and $\sigma_{\mathbf{Y}}^2(t,t^*)$.

### I.1.2 Generalized Ornstein-Uhlenbeck processes

When parameters are allowed to vary over samples, these have to be encoded in the hidden state of GRU-ODE-Bayes, rather than in the fixed weights. For $\mathbf{r}_i$ and $\sigma_i$, GRU-Bayes computes and stores their current estimates as the observations arrive. This is based on previous hidden and current observation as in Eq. 4. The GRU-ODE module then simply has to keep these estimates unchanged between observations:

$$\frac{d\mathbf{r}_i(t)}{dt} = \frac{d\sigma_i(t)}{dt} = 0.$$

This can be easily done by switching off the update gate (*i.e.*, setting $\mathbf{z}(t)$ to 1 for these dimensions). These hidden states can then be used to output the mean and variance in Eq. 6, thus enabling the model to represent generalized Ornstein-Uhlenbeck processes with sample-dependent $\mathbf{r}_i$ and $\sigma_i$.

Perfect representation for sample dependent $\theta_i$ requires the multiplication of inputs in Eq. 6, which GRU-ODE is not able to perform exactly but should be able to approximate reasonably well. If an exact representation is required, the addition of a bilinear layer is sufficient.

Furthermore, the same reasoning applies when parameters are also allowed to change over time within the same sample. GRU-Bayes is again able to update the hidden vector with the new estimates.

### I.1.3 Non-aligned time series

Our approach can also handle samples that would be dephased in time (*i.e*, the observation windows are not aligned on an intrinsic time scale). Longitudinal patient data recorded at different stages of the disease for each patient is one example, developed in Section 5. This setting is naturally handled by the GRU-Bayes module.

## I.2 Case Study: 2D Ornstein-Uhlenbeck Process

### I.2.1 Setup

We evaluate our model on a 2-dimensional OU process with correlated Brownian motion as defined in Eq. 5. For best illustration of its capabilities, we consider the three following cases.

In the first setting, $\mathbf{r}_i$ varies across samples as $\mathbf{r}_i^1 \sim \mathcal{U}(0.5, 1.5)$ and $\mathbf{r}_i^2 \sim \mathcal{U}(-1.5, -0.5)$. The correlation between the Wiener processes $\rho$ is set to 0.99. We also set $\sigma = 0.1$ and $\theta = 1$. The second case, which we call *random lag* is similar to the first one but adds an extra uniformly distributed random lag to each sample. Samples are then time shifted by some $\Delta_T \sim \mathcal{U}(0, 0.5)$. The third setting is identical to the first but with $\rho = 0$ (*i.e.*, both dimensions are independent and no information is shared between them).

We evaluate all methods and settings on the forecast of samples after time $t = 4$. The training set contains 10,000 samples with an average of 20 observations scattered over a 10-second time interval.

Figure 4: Example of predictions (with shaded confidence intervals) given by GRU-ODE-Bayes for two samples of a correlated 2-dimensional stochastic process (dotted line) with unknown parameters. Dots show the observations. Only a few observations are required for the model to infer the parameters. Additionally, GRU-ODE-Bayes learns the correlation between the dimensions resulting in updates of nonobserved variables (red dashed arrow).

Table 4: NegLL and MSE results for 2-dimensional general Ornstein-Uhlenbeck process.

| | NEGATIVE LOG-LIKELIHOOD | | | MSE | | |
|---|---|---|---|---|---|---|
| MODEL | RANDOM $r$ | RANDOM LAG | $\rho = 0$ | RANDOM R | RANDOM LAG | $\rho = 0$ |
| NEURALODE-VAE-MASK | 0.222 | 0.223 | 0.204 | 0.081 | 0.069 | 0.081 |
| NEURALODE-VAE | 0.183 | 0.230 | 0.201 | 0.085 | 0.119 | 0.113 |
| GRU-ODE-BAYES | $-1.260$ | $-1.234$ | $-1.187$ | 0.005 | 0.005 | 0.006 |
| GRU-ODE-BAYES-MINIMAL | $-1.257$ | $-1.226$ | $-1.188$ | 0.005 | 0.006 | 0.006 |
| GRU-ODE-BAYES-SEQ | $-1.266$ | $-1.225$ | $-1.191$ | 0.005 | 0.005 | 0.006 |
| GRU-ODE-BAYES-SEQ-MINIMAL | $-1.260$ | $-1.225$ | $-1.189$ | 0.005 | 0.005 | 0.006 |

Models are trained with a negative log-likelihood objective function, but mean square errors (MSE) are also reported. We compare our methods to NeuralODE-VAE (Chen et al., 2018). Additionally, we consider an extended version of this model where we also feed the observation mask, called NeuralODE-VAE-Mask.

### I.2.2 Empirical evaluation

Figure 1 shows a comparison of predictions between NeuralODE-VAE and GRU-ODE-Bayes for the same sample issued from the random $\mathbf{r}_i$ setting. Compared to NeuralODE-VAE, which retrieves the average dynamics of the sample, our approach detects the correlation between both features and updates its predictions more finely as the observations arrive. In particular, note that GRU-ODE-Bayes updates its prediction and confidence on a feature even when only the other one is observed, taking advantage from the fact that they are correlated. This can be seen on the left pane of Figure 1 where at time $t = 3$ Dimension 1 (blue) is updated because of the observation of Dimension 2 (green).

By directly feeding sporadic inputs into the ODE, GRU-ODE-Bayes sequentially *filters* the hidden state and thus estimates the PDF of the future observations. This is the core strength of the proposed method, allowing it to perform long-term predictions. In contrast, NeuralODE-VAE first stores the whole dynamics in a single vector and later maps it to the dynamics of the time series (illustrated in Figure 1).

This analysis is confirmed by the performance results presented in Table 4. Our approach performs better on all setups for both NegLL and MSE. What is more, the method deals correctly with *lags* (*i.e.*, the second setup) as it results in only marginal degradation of NegLL and MSE. When there is

no correlation between both dimensions (*i.e.,* $\rho = 0$), the observation of one dimension contains no information on the other and this results in lower performance.

Figure 5 illustrates how GRU-ODE-Bayes updates its prediction and confidence as more and more observations are processed. This example is for the first setup (randomized $\mathbf{r}_i$). Initially, the predictions have large confidence intervals and reflect the general statistics of the training data. Then, observations gradually reduce the variance estimate as the model refines its predictions of the parameter $\mathbf{r}_i$. As more data is processed, the predictions converge to the asymptotic distribution of the underlying process.

Figure 5: GRU-ODE-Bayes updating its prediction trajectory with every new observation for the random $\mathbf{r}_i$ setup. Shaded regions are propagated confidence intervals conditioned on previous observations.

## J   Application to synthetic nonlinear SDE: the Brusselator

On top of the extended multivariate OU process, we also studied a nonlinear SDE. We derived it from the Brusselator ODE, which was proposed by Ilya Prigogine to model autocatalytic reactions (Prigogine, 1982). It is a 2-dimensional process characterized by the following equations:

$$\frac{dx}{dt} = 1 + (b+1)x + ax^2 y$$
$$\frac{dy}{dt} = bx - ax^2 y$$

Where $x$ and $y$ stand for the two dimensions of the process and $a$ and $b$ are parameters of the ODE. This system becomes unstable when $b > 1 + a$. We add a stochastic component to this process to make it the following SDE, which we will model:

$$\frac{dx}{dt} = 1 + (b+1)x + ax^2 y + \sigma dW_1(t)$$
$$\frac{dy}{dt} = bx - ax^2 y + \sigma dW_2(t)$$
(7)

Where $dW_1(t)$ and $dW_2(t)$ are correlated Brownian motions with correlation coefficient $\rho$. We simulate 1,000 trajectories driven by the dynamics given in Eq. 7 with parameters $a = 0.3$ and

$b = 1.4$ such that the ODE is unstable. Figure 6 show some realization of this process. The data set we use for training consists in random samples from those trajectories of length 50. We sample sporadically with an average rate of 4 samples every 10 seconds.

Figure 6: Examples of generated trajectories for the stochastic Brusselator Eq. 7 over 50 seconds. Trajectories vary due to stochastic component and sensitivity to initial conditions. Orange and blue lines represent both components of the process.

Figures 7 show the predictions of the trained model on different samples of the proposed stochastic Brusselator process (newly generated samples). At each point in time are displayed the means and the standard deviation of the *filtered* process. We stress that it means that those predictions only use the observations prior to them. Red arrows show that information is shared between both dimensions of the process. The model is able to pick up the correlation between dimensions to update its belief about one dimension when only the other is observed. The model presented in these figures used 50 dimensional latents with DOPRI solver.

Figure 7: Examples of predicted trajectories for the Brusselator. The model has been trained with DOPRI solver. Solid line shows the predicted *filtered* mean, the shaded areas show the 95% confidence interval while dotted lines represent the true generative process. The dots show the available observations for the filtering. Red arrows show the collapse of the belief function from one dimension to another.

# K MIMIC-III: preprocessing details

MIMIC-III is a publicly available database containing deidentified health-related data associated for about 60,000 admissions of patients who stayed in critical care units of the Beth Israel Deaconess

Medical Center between 2001 and 2012. To use the database, researchers must formally request access to the data via http://mimic.physionet.org.

### K.1 Admission/Patient clean-up

We only take a subset of admissions for our analysis. We select them on the following criteria:

- Keep only patient who are in the metavision system.
- Keep only patients with single admission.
- Keep only patients whose admission is longer than 48 hours, but less than 30 days.
- Remove patients younger than 15 years old at admission time.
- Remove patients without chart events data.
- Remove patients with fewer than 50 measurements over the 48 hours. (This corresponds to measuring only half of retained variable a single time in 48 hours.)

This process restricts the data set to 21,250 patients.

### K.2 Variables preprocessing

The subset of 96 variables that we use in our study are shown in Table 5. For each of those, we harmonize the units and drop the uncertain occurrences. We also remove outliers by discarding the measurements outside the 5 standard deviations interval. For models requiring binning of the time series, we map the measurements in 30-minute time bins, which gives 97 bins for 48 hours. When two observations fall in the same bin, they are either averaged or summed depending on the nature of the observation. Using the same taxonomy as in Table 5, lab measurements are averaged, while inputs, outputs, and prescriptions are summed.

This gives a total of 3,082,224 unique measurements across all patients, or an average of 145 measurements per patient over 48 hours.

## L  USHCN-Daily: preprocessing details

The United States Historical Climatology Network (USHCN) data set contains data from 1,218 centers scattered across the US. The data is publicly available and can be downloaded at the following address: https://cdiac.ess-dive.lbl.gov/ftp/ushcn_daily/. All states files contain daily measurements for 5 variables: precipitation, snowfall, snow depth, maximum temperature and minimum temperature.

### L.1 Cleaning and subsampling

We first remove all observations with a bad quality flag, then remove all centers that do not have observation before 1970 and after 2001. We then only keep the observations between 1950 and 2000. We subsample the remaining observations to keep on average 5% of the observations of each center. Lastly, we select the last 4 years of the kept series to be used in the analysis.

This process leads to a data set with 1,114 centers, and a total of 386,068 unique observations (or an average of 346 observations per center, sporadically spread over 4 years).

## M  Small-sample regime: additional results

In the main text of the paper, we presented the results for the Mean Square Error (MSE) for the different data subsets of MIMIC. In Table 6, we present the negative log-likelihood results. They further illustrate that the continuity prior embedded in our GRU-ODE-Bayes strongly helps in the small-sample regime.

| Retained Features | | | |
|---|---|---|---|
| Lab measurements | Inputs | Outputs | Prescriptions |
| Anion Gap | Potassium Chloride | Stool Out Stool | D5W |
| Bicarbonate | Calcium Gluconate | Urine Out Incontinent | Docusate Sodium |
| Calcium, Total | Insulin - Regular | Ultrafiltrate Ultrafiltrate | Magnesium Sulfate |
| Chloride | Heparin Sodium | Gastric Gastric Tube | Potassium Chloride |
| Glucose | K Phos | Foley | Bisacodyl |
| Magnesium | Sterile Water | Void | Humulin-R Insulin |
| Phosphate | Gastric Meds | TF Residual | Aspirin |
| Potassium | GT Flush | Pre-Admission | Sodium Chloride 0.9% Flush |
| Sodium | LR | Chest Tube 1 | Metoprolol Tartrate |
| Alkaline Phosphatase | Furosemide (Lasix) | OR EBL | |
| Asparate Aminotransferase | Solution | Chest Tube 2 | |
| Bilirubin, Total | Hydralazine | Fecal Bag | |
| Urea Nitrogen | Midazolam (Versed) | Jackson Pratt 1 | |
| Basophils | Lorazepam (Ativan) | Condom Cath | |
| Eosinophils | PO Intake | | |
| Hematocrit | Insulin - Humalog | | |
| Hemoglobin | OR Crystalloid Intake | | |
| Lymphocytes | Morphine Sulfate | | |
| MCH | D5 1/2NS | | |
| MCHC | Insulin - Glargine | | |
| MCV | Metoprolol | | |
| Monocytes | OR Cell Saver Intake | | |
| Neutrophils | Dextrose 5% | | |
| Platelet Count | Norepinephrine | | |
| RDW | Piggyback | | |
| Red Blood Cells | Packed Red Blood Cells | | |
| White Blood Cells | Phenylephrine | | |
| PTT | Albumin 5% | | |
| Base Excess | Nitroglycerin | | |
| Calculated Total CO2 | KCL (Bolus) | | |
| Lactate | Magnesium Sulfate (Bolus) | | |
| pCO2 | | | |
| pH | | | |
| pO2 | | | |
| PT | | | |
| Alanine Aminotransferase | | | |
| Albumin | | | |
| Specific Gravity | | | |

Table 5: Retained longitudinal features in the intensive care case study.

Table 6: Vitals forecasting results on MIMIC-III (NegLL) - Low number of samples setting

| | 1,000 PATIENTS | 2,000 PATIENTS | FULL |
|---|---|---|---|
| MODEL | NegLL | NegLL | NegLL |
| NEURAL-ODE | $1.40 \pm 0.01$ | $1.39 \pm 0.005$ | $1.35 \pm 0.01$ |
| GRU-DISC-BAYES | $1.35 \pm 0.01$ | $1.20 \pm 0.015$ | $\mathbf{0.74} \pm 0.04$ |
| GRU-ODE-BAYES | $\mathbf{1.23} \pm 0.006$ | $\mathbf{1.13} \pm 0.01$ | $0.83 \pm 0.04$ |

# N   Computing Infrastructure

All models were run using a NVIDIA P100 GPU with 16GB RAM and 9 CPU cores (Intel(R) Xeon(R) Gold 6140). Implementation was done in Python, using Pytorch as autodifferentitation package. Required packages are available in the code

# O   Hyper-parameters used

All methods were trained using the same dimension for the hidden $\mathbf{h}$, for sake of fairness. For each fold, we tuned the following hyper-parameters using a 20% left out validation set:

**Dropout** rate of 0, 0.1, 0.2 and 0.3.

**Weight decay**: 0.1, 0.03, 0.01, 0.003, 0.001, 0.0001 and 0.

**Learning rate** : 0.001 and 0.0001

Best model was selected using early stopping and performance were assessed by applying the best model on a held out test set (10% of the total data). The different folds were reused for each compared model for sake of reproducibility and fair comparison. We performed 5-fold cross validation and present the test performance average and standard deviation in all tables.