[Reviews · NeurIPS 2019]

Reviewer 1



Originality: In this paper, the authors propose the GRU-ODE-Bayes method to continuously model the sporadically-observed time series. The proposed method is a novel combination of the GRU-ODE and the GRU-Bayes. The authors change the classical GRU into the difference equation version so that it can deal with continuous time. Moreover, the proposed GRU-Bayes can handle sporadical observations. Clarity: The paper is well written. And it would be better if the authors can put Section 3 related research just after the introduction section so that the readers can have a better understanding of the field after reading the introduction. Significance: The authors show the experiment on both synthetic and real-world datasets and the experimental results of the proposed method are much better than the baseline methods. # Feedback The authors' feedback solved some of my concerns.

Reviewer 2



Summary: This paper proposes a method for modelling sporadically measured multivariate time-series. The approach builds off the Neural ODE-VAE model (Chen, 2018, section 5) which does forecasting by solving an ODE based on an initial state, which is given by a representation of the time-series up to that point (e.g. the output of a RNN encoder). In this paper, they first propose a continuous-time version of a gated recurrent unit (GRU-ODE) which produces an ODE for how the hidden state of a recurrent network should evolve over time. They additionally describe how to update the hidden state of this RNN (GRU-Bayes) and propose a composite model (GRU-ODE-Bayes). The approach is explored on synthetic data and compared to existing models on EHR and climate data. Overall impression: I think this work describes an interesting approach to an enduring challenge in time-series modelling (sporadic sampling). It appears to work quite well in the tested domains. Although I have some reservations about some of the experiments, I do not think they constitute fatal flaws of the approach, rather may limit the significance of the work. I have a confusion about the model itself (see 2b) which I hope the authors can clarify. 2.a Originality: ["Are the tasks or methods new? Is the work a novel combination of well-known techniques? Is it clear how this work differs from previous contributions? Is related work adequately cited?"] While the work builds naturally on Chen 2018, the formulation of GRU in terms of an ODE appears to be novel, and the method otherwise is an evolution over the earlier approach. The related work is quite comprehensively covered, although there is also a notable line of work using point process-based models (e.g. Mei & Eisner, NeurIPs 2017) which also target sporadically sampled time-series. Possibly contemporaneously, work from Chang, Chen, Haber and Chi at ICLR 2019 explores RNNs as dynamical systems, which touches on the GRU-ODE part of this work. 2.b Quality: ["Is the submission technically sound? Are claims well supported by theoretical analysis or experimental results? Is this a complete piece of work or work in progress? Are the authors careful and honest about evaluating both the strengths and weaknesses of their work?"] The submission is overall reasonably sound, although I have some comments and questions: * Regarding the model itself, I am confused by the GRU-Bayes component. I must be missing something, but why is it not possible to ingest observed data using the GRU itself, as in equation 2? This confusion would perhaps be clarified by an explanation in line 89 of why continuous observations are required. As it is written, I am not sure why it you couldn't just forecast (by solving the ODE defined by equation 3) the hidden state until the next measurement arrives, at which point g(t) and z(t) can be updated to define a new evolution equation for the hidden state. I am guessing the issue here is that this update only changes the derivative of the hidden state and not its value itself, but since the absolute value of the hidden state is not necessarily meaningful, the problem with this approach isn't very clear to me. I imagine the authors have considered such a model, so I would like to understand why it wouldn't be feasible here. * In lines 143-156, it is mentioned that the KL term of the loss can be computed empirically for binomial and Gaussian distributions. I understand that in the case of an Ornstein-Uhlenbeck SDE, the distribution of the observations are known to be (conditionally) Gaussian, but in the case of arbitrary data (e.g. health data), as far as I'm aware, few assumptions can be made of the underlying process. In this case, how is the KL term managed? Is a Gaussian distribution assumption made? Line 291 indicates this is the case, but it should be made clear that this is an assumption imposed on the data. For example, in the case of lab test results as in MIMIC, these values are rarely Gaussian-distributed and may not have Gaussian-distributed observation noise. On a similar note, it's mentioned in line 154 that many real-world cases have very little observation noise relative to the predicted distribution - I assume this is because the predicted distribution has high variance, but this statement could be better qualified (e.g. which real-world cases?). * It is mentioned several times (lines 203, 215) that the GRU (and by extension GRU-ODE-Bayes) excels at long-term forecasting problems, however in both experiments (sections 5.2 and 5.3) only near-term forecasting is explored - in both cases only the next 3 observations are predicted. To support this claim, longer prediction horizons should be considered. * I find it interesting that the experiments on MIMIC do not use any regularly-measured vital signs. I assume this was done to increase the "sporadicity" of the data, but it makes the application setting very unrealistic. It would be very unusual for values such as heart rate, respiratory rate, blood pressure and temperature not to be available in a forecasting problem in the ICU. I also think it's a missed opportunity to potentially highlight the ability of the proposed model to use the relationship between the time series to refine the hidden state. I would like to know why these variables were left out, and ideally how the model would perform in their presence. * I think the experiment in Section 5.5 is quite interesting, but I think a more direct test of the "continuity prior" would be to explicitly test how the model performs (in the low v. high data cases) on data which is explicitly continuous and *not* continuous (or at least, not 2-Lipschitz). The hypothesis that this continuity prior is useful *because* it encodes prior information about the data would be more directly tested by such a setup. At present, we can see that the model outperforms the discretised version in the low data regime, but I fear this discretisation process may introduce other factors which could explain this difference. It is slightly hard to evaluate because I'm not entirely sure what the discretised version consists of , however - this should be explained (perhaps in the appendix). Furthermore, at present there is no particular reason to believe that the data in MIMIC *is* Lipschitz-2 - indeed, in the case of inputs and outputs (Table 4, Appendix), many of these values can be quite non-smooth (e.g. a patient receiving aspirin). * It is mentioned (lines 240-242, section H.1.3) that this approach can handle "non-aligned" time series well. As mentioned, this is quite a challenging problem in the healthcare setting, so I read this with some interest. Do these statements imply that this ability is unique to GRU-ODE-Bayes, and is there a way to experimentally test this claim? My intuition is that any latent-variable model could in theory capture the unobserved "stage" of a patient's disease process, but if GRU-ODE-Bayes has some unique advantage in this setting it would be a valuable contribution. At present it is not clearly demonstrated - the superior performance shown in Table 1 could arise from any number of differences between this model and the baselines. 2.c Clarity: ["Is the submission clearly written? Is it well organized? (If not, please make constructive suggestions for improving its clarity.) Does it adequately inform the reader? (Note: a superbly written paper provides enough information for an expert reader to reproduce its results.)"] While I quite like the layout of the paper (specifically placing related work after a description of the methodology, which is somewhat unusual but makes sense here) and think it is overall well written, I have some minor comments: * Section 4 is placed quite far away from the Figure it refers to (Figure 1). I realise this is because Figure 1 is mentioned in the introduction of the paper, but it makes section 4 somewhat hard to follow. A possible solution would be to place section 4 before the related research, since the only related work it draws on is the NeuralODE-VAE, which is already mentioned in the Introduction. * I appreciate the clear description of baseline methods in Section 5.1. * The comprehensive Appendix is appreciated to provide additional detail about parts of the paper. I did not carefully read additional experiments described in the Appendix (e.g. the Brusselator) out of time consideration. * How are negative log-likelihoods computed for non-probabilistic models in this paper? * Typo on line 426 ("me" instead of "we"). * It would help if the form of p was described somewhere near line 135. As per my above comment, I assume it is a Gaussian distribution, but it's not explicitly stated. 2.d Significance: ["Are the results important? Are others (researchers or practitioners) likely to use the ideas or build on them? Does the submission address a difficult task in a better way than previous work? Does it advance the state of the art in a demonstrable way? Does it provide unique data, unique conclusions about existing data, or a unique theoretical or experimental approach?"] This paper describes quite an interesting approach to the modelling of sporadically-measured time series. I think this will be of interest to the community, and appears to advance state of the art even if it is not explicitly clear where these gains come from.

Reviewer 3



ORIGINALITY: I will caveat my review by noting that I have not followed the NODE-related literature closely, although I am quite familiar with RNN applications to clinical time series. A Google Scholar search through papers citing the Chen 2018 paper indicate that although it has been cited over a hundred times, there has been very little directly related follow-up work. In particular, as far as I can tell, there is little-to-no previous work anticipating this paper's use of a GRU cell to parameterize the derivative of the hidden state, nor its integration with Bayesian filtering. What is more, I am not aware of any prior attempts to apply NODE-like architectures to climate or health time series, both ideal applications for this sort of model. Hence, I consider this work to be very original. QUALITY: Overall, the work appears well done, and I am unable to detect any obvious flaws or weaknesses in the proposed methods themselves (once again, I will note that while I know RNNs quite well, I am rusty on ODEs and only superficially familiar with recent neural ODE work). Most of the questions I have are more related to how the proposed approach is motivated and described, which I will address under Clarity. The experimental design seems sound for the most part; it is perhaps a little frustrating that critical experimental design details are sometimes deferred to the appendix, but this is unavoidable given the NeurIPS page limit, and ultimately, it is not a barrier to reproducibility -- especially since the authors use public datasets and have made the code available. One relatively big concern I have is the train/test split procedure for each experiment and how hyperparameters were tuned. The only mention of a test set is buried in lines 592-594 of the appendix in this ominous sentence: "Best model was selected using early stopping and performance were assessed by applying the best model on a held out test set (10% of the total data). We performed 5-fold cross validation and present the test performance average and standard deviation in all tables." A straightforward read suggests that k-fold cross validation with five different test sets was used instead of a fully independent test set and hyperparameters were tuned on test set performance (!). I'd like the authors to address this and the below questions in the response and to make it crystal clear in the MAIN BODY of the manuscript: * Am I correct that they used five-fold cross validation with no fully independent test set? * Are the folds fixed across runs or generated randomly at the start of each run? * Assuming so, can the authors estimate how many times they examined test set performance (or individual test examples) during experimentation before reporting their final numbers? With this number on hand, can they discuss the risk of test set leakage (direct or indirect) and steps that they took (or could take, in the future) to reduce this risk? Can they also discuss the pros and cons of this approach vs. having an independent test set that they use only at the very end? * How exactly did the authors tune hyperparameters? During each k-fold split, do they set aside some data as a validation set, or are they tuning on test set accuracy? * Did the authors invest the same amount of effort in tuning the hyperparameters of baselines as they did the proposed model? CLARITY: The presentation of the idea and experiments could use some work. As noted above in Quality, some critical aspects of experimental design were omitted. It is not necessary for the reader to be able to reproduce exactly the methods and experiments from just reading the main body -- especially if the details are clearly stated in the appendix and code is shared -- but things like, e.g., train/test procedure split are essential for the reader to be able to interpret the results. As for the proposed methods themselves, the GRU-Bayes component did not feel clearly motivated. Admittedly, my experience with RNNs has focused on predicting an independent target, e.g., classification, vs. next step ahead forecasting, but the RNN forecasting work I've seen I think usually uses a simpler input-output problem formulation, akin to language modeling, without casting it as Bayesian filtering. Is this a necessary component for adding the ODE piece, or is it just an additional improvement? On that note, if the ODE and Bayes components are independent, I'd love to see an additional row in Table 2 for a plain "GRU-ODE." Another detail: my understanding from the Chen paper is that minibataching with the ODE architecture is non-trivial. I didn't fully understand the description on lines 159-162 of how the authors dealt with these challenges. Also, it's not clear to me at all (from this paper or Chen) how backpropagation-through-time works in this setting. Since this paper is an extension of the NODE idea to a recurrent setting, I think this would be a very valuable discussion to include in the paper's main body. As far as I can tell, the only prior work on neural ODEs that the paper cites is the Chen paper, but a quick Google Scholar search and some citation following yields a number of papers published on the topic over the last year, including several by Ruthotto and Haber. Can the authors place their work in the larger context of research on the intersection of neural nets and ODEs, not just the Chen paper? Also, is there any connection to continuous time Bayes nets or piecewise-constant conditional intensity models? SIGNIFICANCE: Setting aside my concerns about the risk of test set leakage, this paper seems like an important advance in time series modeling with RNNs. Continuous time dynamics and irregular sampling present fundamental challenges in the field, particularly in key application areas like healthcare, and solutions that rely on discretization and imputation or heuristics (like passing in time deltas) feel unsatisfactory. Ideas from dynamical systems and control have been under-explored, at least in deep learning, and seem to offer very promising directions of inquiry. This paper seems like a nice (albeit arguably incremental) extension of NODE to recurrent architectures and a valuable empirical demonstration of its effectiveness for real world time series modeling. My suspicion is that in the big picture, this paper will be dwarfed by previous work on neural ODEs, but it is nonetheless valuable follow-up work and should be of interest to practitioners.

Reviewer 4



The paper is well written and presents an interesting approach to a current and interesting problem. It overcomes some of the strangeness in NeuralODE in which the network performs a strange combination of smoothing and filtering. While I am overall positive on this paper, I think it could be improved in a few ways: 1. It might help readers to have a bit more discussion of the relationship of this work to other different, but related problems. For instance, this paper assumes that the observation times are selected irrespective of the state of the system. Other papers model the intensity of the observation times (see the papers on point processes). Both are useful, but to distinguish this work, it would be helpful for many readers to contrast the approaches. Similarly, there are works on filtering and estimating the parameters of SDEs. This paper does not follow that line, instead encouraging (but not forcing) the NN to approximate Bayesian updates of a SDE. Especially given that the first equation is for a SDE (which is never really referenced again), making this distinction (between SDE inference and this approach) would be helpful (perhaps with a reference to something like Sarkka and Solin's text on Applied SDEs?). 2. It is not clear how much of the performance is due to the use of an "ODE GRU" and how much is due to objective function. I would have liked to see the GRU replaced with a basic MLP to see the difference. 3. While the emperical evaluations involved a couple of real-world datasets, the criteria used might not be the most relevant for those particular domains. For instance, in the MIMIC III dataset, there are established benchmarks (see Harutyunyan et al., Multitask Learning and Benchmarking with Clinical Time Series Data, 2019, for instance). If not using those, I would have preferred to see llh on testing data, rather than predicting the next three observations. Yet, overall, this is a nice paper. The experimental results are the weakest point, but for a paper focused on introducing a new method for Bayesian filtering of continuous-time processes, this is primarily an algorithms/architecture paper.

[Author Response · NeurIPS 2019]

We thank reviewers for the relevant comments. We first address general questions and then give brief individual answers.

**On the necessity of GRU-Bayes and link to filtering methods.** GRU-ODE and GRU-Bayes have complementary roles and should be used together. GRU-ODE integrates the dynamics of the hidden process $h(t)$ in time. It thus provides the future estimated distribution of the observations. It computes our **belief** about the unobserved future time series observations. Those projected distributions vary smoothly as they are driven by an ODE. But, as soon as any new observation is available, our belief about the state of the process should be updated **instantaneously**. GRU-Bayes is responsible for this and compares the prediction from GRU-ODE and the actual observation. This update necessarily has to be discrete to accommodate for the new information arriving in packets (or sporadically). See figures 1 and 2 for illustration of this dynamic. This also motivates why sporadic measurements cannot be fed directly to the ODE : observing a sample should make us update our belief immediately. A useful analogy is the Kalman filter, which consists in a **prediction** and an **update** phase. In prediction phase, it propagates in time the predictions about the distribution of the state (analog to GRU-ODE). In update phase, it computes a new estimate for the state probability distribution conditioned on the new information (analog to GRU-Bayes). Yet, in contrast to the (extended) Kalman filter, our approach is able to learn complex non-linear dynamics for the hidden process and is computationally cheaper.

**More related works context.** Other recent works have previously investigated the relationship between deep neural nets and differential equations. They mainly focused on deriving better deep architectures motivated by the stability of the corresponding partial differential equations (Ruthotto and Haber, *arXiv* 2018; Chang et al., *ICLR* 2019). Despite their ODE motivation, those approaches aim at designing new **discrete** architectures and don't explore neural networks parametrized ODEs as such.

**Point processes** (Mei and Eisner, *NeurIPS* 2017; Gunawardana et al., *NeurIPS* 2011) are intrinsically continuous as they focus on **time-to-event** modelling. Continuous-time Bayesian networks (Nodelman et al., *UAI* 2002) address a related problem where they frame events as state transitions. In contrast, our work aims at modelling continuous-time **real-valued** measurements, not only events. Yet, our continuous modelling of the latent process $h(t)$ allows to easily **jointly** model a continuous intensity function (*e.g.* by modulating the intensity function of a Poisson process with $h(t)$). This joint modeling of continuous measurements and events was left for future work.

**Reviewer 2.** Some assumptions have to be made about the conditional distribution of the observations. However, this is not very restrictive as a broad range of distributions have a tractable KL (*e.g.* Poisson, Exponential, ...).

The main difference with GRU-Discretized-Bayes resides in the continuity of the latent process (will be detailed in the Appendix). We then considered this experiment as a continuity ablation study. Note that the continuity prior can be tuned by rescaling time accordingly. Elongating time by a factor 2 would lead to a Lipschitz-4 prior.

Unlike sporadic measurements, the continuous measurements can indeed be directly fed to the GRU-ODE as suggested at line 89 and would considerably improve the prediction of the model as vital signs are very strong predictors in critical care. In this work we didn't include them (1) to compare equally with the other baselines which are not capable of handling continuous inputs, (2) we re-use the same variable subsets as used in (Che et al. *Scientific Reports* 2018). Still, we warmly welcome this suggestion from the reviewer as this would further highlight the capabilities of our model.

For fair comparison, all compared models were *made* probabilistic if not already (*i.e.* they all output log-variance).

**Reviewer 4.** We added the following (at l.264) : "We assessed the performance of the models by creating 5 different folds, each consisting of training (70%), validation (20%) and a left out test set (10%). Those folds are **reused** across compared architectures. For each fold, the models were trained with various hyper-parameters on the train set then evaluated on the validation set to select the best ones. We then retrain the model on the train data and assess performance on the test set. We do this 5 times and report the mean and stdev of our test set performances." Test samples were **never** used in training nor in tuning hyper-parameters, but for performance reporting **only**. The same train, val and test indices were used to compare all models and we used the **same number of hyper-parameters combinations** for all methods.

Minibatch is performed by jointly integrating all time series in the batch between unique observation times over the whole batch. At each unique time, we only update (with GRU-Bayes) the time series actually having an observation at that time and leave the other untouched. We then proceed with joint time integration until the next unique time.

**Reviewer 5.** A basic MLP does not enjoy the properties stated in sections 2.2 and in first paragraph of section 2.3. Still we investigated an other ODE parametrization : *GRU-minimal*, as described in Appendix G.

A important strength of our approach is to be able to predict future observations at **any** point in time. Yet it can also be easily used for time series classification or regression (e.g. using the last latent $h(T)$ to feed a classifier). If those labels are widely used in the literature, we are convinced vitals forecasting is just as important in medical practice as the health status of a patient is complex and cannot usually be fully captured by discrete labels.

[Meta-Review · NeurIPS 2019]

This paper proposes a method for modeling irregularly sampled multivariate time-series using a novel continuous-time version of a gated recurrent unit (GRU-ODE) to parameterize the gradient of the latent state combined with a probabilistic model for accommodating discontinuties in the latent state due to encountering new observations (GRU-Bayes). The problem the authors are solving is important and the method proposed in novel. The author response addressed a number of reviewer questions, but some important items remained. The most import is that several of the reviewers would like to see a number of ablations to the model to provide better evidence for where the improved performance is deriving from. Several reviewers mentioned wanting to see results when the GRU-Bayes part of the model is removed and only the GRU-ODE part of the model is used. This will help to quantify the importance of including the GRU-Bayes portion of the model. Several reviewers also asked for a broader review of the related literature to better situate the work. Nevertheless, the consensus opinion of the reviewers is that the paper is over the bar for acceptance. The authors should carefully take the items mentioned in this meta review into account when updating the manuscript, as well as the suggestions provided in the individual reviews.